# Noise Exposure and Hearing Loss among Brewery Workers in Lagos, Nigeria

**DOI:** 10.3390/ijerph17082880

**Published:** 2020-04-22

**Authors:** Nino L. Wouters, Charlotte I. Kaanen, Petronella J. den Ouden, Herbert Schilthuis, Stefan Böhringer, Bas Sorgdrager, Richard Ajayi, Jan A. P. M. de Laat

**Affiliations:** 1Clinical Technology, Delft University of Technology, 2628 CD Delft, The Netherlands; charlottekaanen@live.nl; 2Global Health, HEINEKEN International B.V. Amsterdam, 1017 ZD Amsterdam, The Netherlands; ellen.denouden@heineken.com (P.J.d.O.); herbert.schilthuis@heineken.com (H.S.); 3Biomedical Data Sciences, Leiden University Medical Centre, 2333 ZA Leiden, The Netherlands; s.boehringer@lumc.nl; 4Coronel Institute of Occupational Health, Amsterdam University Medical Centre, 1105 AZ Amsterdam, The Netherlands; sorgdrager@basbgz.nl; 5Company Clinic, Nigerian Breweries Plc. Iganmu House, Abebe Village Rd, Iganmu, Lagos 101271, Nigeria; richard.ajayi@heineken.com; 6Department of Audiology (ENT), Leiden University Medical Centre, 2333 ZA Leiden, The Netherlands; J.A.P.M.de_Laat@lumc.nl

**Keywords:** occupational health, audiology, noise hazards, NIHL, cross-sectional study, personal noise dosimetry, Nigeria, Sub-Saharan Africa

## Abstract

The health risks of exposure to loud noises are a well-established fact and are widely addressed in modern industries. Yet, in less developed countries, it is thought these hazards receive less attention, both in the workplace and in private life. *(1) Background*: The aim of this study is to assess the occupational noise exposure in a developing country and identify possible risk groups for whom intervention is needed. *(2) Methods*: A cross-sectional study was performed among brewery employees in Lagos, Nigeria. Pure-tone audiometry (PTA) was performed, paired with a self-report questionnaire. Personal noise dosimetry (PND) was also performed with an additional group of participants. *(3) Results*: A total of 458 employees were submitted to PTA. The Packaging and Utilities department reported the largest shifts in hearing thresholds (18 dB [sd = 15] and 16 dB [sd = 15] @4kHz, respectively). No significant effect of department type on auditory health could be found. PND results were obtained from 39 employees. Packaging and Sales were identified as the most exposed departments. *(4) Conclusions*: A healthy hearing profile was found for a large proportion of the brewery employees (91.7%). However, NIHL (noise-induced hearing loss) proportions specifically among Bottling and Sales employees were elevated.

## 1. Introduction

Occupational hazards are a widespread concern in many industries, including the subtopic of noise hazards. Noise exposure brings about noise-induced hearing loss (NIHL) and consequent issues [1,2]. Extensive knowledge on this subject is available in developed countries [3,4,5]. Many industrial hazards known from production plants in these countries, including noise, are already being addressed in developed countries. However, in general, occupational health and safety are thought to receive less attention and be addressed less thoroughly in developing countries. Compliance with safety recommendations may vary across industries and employee self-awareness of exposure may be insufficient [6,7,8]. Specifically, few researches can be found on the subject of NIHL in sub-Saharan Africa. The little information comes from industries such as mining [9] and power plants [10]. Both of these domains report alarming proportions of noise-induced hearing loss. Musiba et al. [9] reported a proportion of NIHL as high as 47% within a population of Tanzanian miners. This is paired with a lack of hearing conservation programs (HCP) as reported by Moroe et al. [11] Nigeria is a rapidly developing country with growing industries where occupational hazards are a widely unknown subject [11]. A previous prospective study from 2008 by Ologe et al. [6] showed a proportion of sensorineural hearing loss of 64.9% in 2003 and 86.9% in 2005. Since then, however, industries have evolved in Nigeria as reflected by a GDP (gross domestic product) that has doubled between 2005 and 2017 (source: www.worldbank.org). A new insight into the field of noise control in the workplace is needed. The objective of this study was to gain insight into the auditory health of brewery employees in a developing country and their exposure to noise at work and in private in order to identify possible groups at risk and key subject areas on which to intervene. This will be attained through two different approaches of the population. At first, audiometric threshold data from a representative sample of the brewing company employees would give insight into the prevalence of NIHL. Secondly, personal dosimetry needed to give information about the possible factors and moments of noise exposure. In the following paper, we start by describing an extensive dataset obtained in a brewery in Nigeria. Next, we present data analyses and possible conclusions.

## 2. Materials and Methods

As expressed earlier, two observational studies are described in this paper. At first, audiometry is reported for it gives the best insight into the population that was observed and that was also subject in the second part: personal dosimetry.

### 2.1. Sample

A cross-sectional study was performed to assess employees’ hearing thresholds. Brewery employees (approximately 1500, both full-time employees and third-party contractors) were categorized by department. For this study, seven departments were distinguished as follows: (a) Packaging, of which Bottling and Canning were considered separately; (b) Utilities, (c) Office, (d) Sales, (e) Brewing, comprising all employees working in and around the brewhouse; and (f) Warehousing, comprising all employees working in shipment and logistics. Randomly selected employees from the full-time employee register were invited to attend the program; an original sample of 250 employees was based on sample size calculations. After completing the informed consent form, subject’s data were anonymized. Participants were given a questionnaire with demographic, employment, and health-related questions. Before the audiometry, otoscopy and tympanometry were performed to identify any abnormalities by two trained students in Clinical Technology supervised by local company doctors. Exclusion criteria from this study were: (a) any middle ear defects, (b) any known hearing defect not due to noise exposure (i.e., pierced eardrum, infections, etc.); and (c) not being a brewing company employee (the clinic was also accessible for employees’ relatives). Results were obtained from 520 participants, adjusted to 458 after exclusion. The analysis was performed using SPSS packages 24 and 25 (IBM Corp., Armonk, NY, USA), focused on the 4 kHz threshold and its surrounding frequencies (1, 2, and 6 kHz) as a measure for NIHL. The resulting threshold values for these frequencies were used as a measure for hearing loss and combined with the questionnaire results through further analysis using ANOVA and frequency reports. As the point of interest was the relationship between departments and possible noise hazards, tests were performed to compare the respective department of an employee with his hearing profile. A random subsample of employees from selected departments was invited to take part in a dosimetry study. Given the variation in activity types, particular interest was taken in the data that could be obtained from the Sales department. Thus, Sales employees were oversampled compared to other departments. 

### 2.2. Audiometry

Pure-tone audiometry was finally performed in a noise-insulating booth (measured levels inside booth were between 33 and 38 dB) at the brewery clinic. Air conduction thresholds were measured on calibrated audiometers (Amplivox^®^ 170, Amplivox Ltd., Birmingham, UK) for both ears at frequencies 500 Hz, 1 kHz, 2 kHz, 4 kHz, 6 kHz, and 8 kHz.

### 2.3. Personal Dosimetry

Subjects were required to wear a personal dosimeter (Etymötic ER-200DW8, Etymötic Research Inc., Elk Grove Village, IL, USA) for 24 h while carrying out their daily routine. During the recordings, participants were asked to keep an hourly log of their whereabouts. The recording threshold for the dosimeter was 70 dB, and the value of 35 dB was used to impute the values below that threshold.

## 3. Results

### 3.1. Study Population

As previously stated, results were obtained from 520 participants, 458 after exclusion. Due to logistical difficulties, the original sample of 250 was abandoned and oversampling was done in order to maintain statistical power. Mean age was 35 ± 7.38 years with first employment age at 24 ± 4.04 years. Employees worked for 51.0 ± 15.95 h per week. Some extremely high values were reported (maximum = 144 h/week) with 5% above 84 h. It was not possible to find grounds on which to reject these values as outliers due to the self-reporting nature of the data; however, values above maximum (24 × 7 = 168 h) were considered outliers and removed from the analysis. As this variable is not used in any analysis for this article, the influence of this reporting method was not investigated.

Most of the employees were working in the Packaging department at the Bottling line (*n* = 186, 40.6%, see Table 1). Data were collected from a large sample of people from the Sales (*n* = 101, 22%) and Office departments (*n* = 88, 19.2%) as well. In the far-right column of Table 1, estimates of the workforce as of December 2019 (study was performed two years prior) are shown. These estimates were obtained in the context of a status update from the brewery after the study took place. Table 2 illustrates the age distribution of workers, gender included. More male (*n* = 399) than female (*n* = 58) employees attended the program as reported by the participants (1 subject did not report gender). Most (*n* = 364) stated that they attended post-secondary education. This included advanced courses or even university degrees.

### 3.2. Audiometric Results

Hearing thresholds recordings were mapped to the reported department of the participant (Figure 1). Simple unpaired T-tests were used on the 4 kHz value to compare each department with Office employees who formed the control group as these were thought to be less exposed to noise. The Bottling department showed a significantly higher hearing level at the 4 kHz level for both right and left ear (*p* = 0.007 and *p* = 0.024, respectively). No other department showed significantly higher hearing levels compared to the profile of Office employees. Furthermore, thresholds for all populations are within the acceptable healthy range (between 0 and 25 dB). A clear association of incidence of age on the hearing threshold was observed, as shown in Figure 2.

To obtain a more detailed impression of the population’s auditive health, BIAP 02/1: Audiometric Classification of Hearing Impairments [12] was used. This classification is the norm for Occupational Health and Safety studies. Hearing thresholds at 500 Hz, 1 kHz, 2 kHz, and 4 kHz were averaged and classified according to the BIAP recommendations. No profound hearing loss was found according to this classification. With increasing age, a larger proportion of mild hearing losses appear, which is a normal phenomenon corresponding to regular presbycusis values. Furthermore, the 4kHz hearing threshold was compared to the three surrounding frequencies (1 kHz, 2 kHz and 6 kHz) to determine net loss at this frequency (Table 3).

In an attempt to find employees at risk for developing NIHL, the 4 kHz hearing threshold was compared to the levels of ambient frequencies. Univariate analysis of variance (ANOVA) was conducted to compare the main effects of the currently occupied department, gender, and age on this difference; results can be seen in Table 4. The results of this analysis could indicate the influence of parameters on the evolution of a variable. In this case, the ANOVA can give insight into the influence of (e.g.) department on an individual in developing NIHL. A positive effect indicates an increase in the hearing loss, while a negative effect indicates a “protective” effect compared to the control population. Reported age was used rather than age groups; the values shown in the ANOVA, therefore, are the result of linear regression on this variable. The main effect for current position was significant (*p* = 0.014, F(6.447) = 193.562), indicating an effect of workplace (current department of employment). Unsurprisingly, the main effect for age was significant as well (*p* < 0.001, F(1.447) = 14.659) because age is a well-known risk factor for hearing loss. The main effect for gender was not significant (*p* = 0.201, F(1.447) = 1.637). Office levels were used as a control group for the ANOVA as it was expected that this group had less exposure to noise. Working in the bottling department showed a significant influence on NIHL development (B = 3.654; *p* = 0.007). Bottling employees were more likely to develop NIHL than Office counterparts. Compared to the Office population, Bottling, Utilities, Sales, and Brewing reported a strong yet nonsignificant positive effect on developing NIHL. While Canning and Warehousing employees showed a negative effect (B = −2.265 and B = −0.407, respectively) compared to Office personnel, this translates to the fact that canning and warehousing employees presented a better hearing profile. However, these differences were not statistically significant either.

A multiple regression (Table 5) test was also performed to evaluate the overall influence of the departments on an individual subject’s risks of developing NIHL. Participants had indicated the previous department (if any) occupied within the brewery and duration. Again, Office was used as the reference group. ANOVA tested a significant combined effect in model 2 for all departments together (*p* < 0.001). Combined with gender and age, the overall time spent working at each department by a single employee had a significant effect on hearing threshold. Only the Bottling department showed a negative influence on hearing thresholds with a regression coefficient of 0.077 (*p* = 0.456). Canning showed a negative coefficient of −0.423 (*p* = 0.057). It was, therefore, not possible to attribute an effect on hearing threshold to a specific department; moreover, the combination with the factors of age and gender makes confounding impossible.

### 3.3. Dosimetry

Dosimetry results were obtained from 39 participants. Specific demographics for this group were not recorded. This part resulted in 931 h of noise recordings. Due to incomplete or incorrect logging by the participants, about 72 h of this data was incomplete and, therefore, excluded from the analysis. Levels were recorded in dB Leq with frequency weighting A. Results are shown in Figure 3 and Figure 4. Utilities and Packaging employees recorded the highest levels with 61.2 (59.4–63.0) dB and 60.9 (60.1–61.7) dB, respectively. These values are far above other departments, even considering the bounds of the 95% confidence interval Sales employees come second in terms of recorded noise levels with an average of 54.4 (54.0–54.9) dB, followed closely by Clinic employees reporting an average ambient noise of 51.8 (51.0–52.5) dB. Both Office and Brewing staff recorded levels under 50 dB (47.7 dB and 49.7 dB, respectively). Looking at the average noise level per location of recording, displayed in Figure 4, there is a bigger variation. Packaging was the loudest location recorded with an average noise level of 75.5 (74.6–76.5) dB. This is about 7 dB more without overlapping CIs than the noise measured at different social events, which is the second loudest place listed with 68.5 (67.5–69.6) dB on average. The hours recorded at the Warehouse report a noise level as high as 66.8 (65.6–67.9) dB and, in the same category, the Brewery Canteen is reported with an average noise level of 65.5 (63.5–67.5) dB. At Utilities, the average noise level recorded was 55.9 (53.8–58.1) dB. Most other locations around the brewery present background noise ranging between 40.5 (39.1–41.9) dB on the Brewery and 59.9 (56.5–63.3) dB at the Brewhouse. At home, an average level of 41.0 (40.7–41.3) dB was recorded, making this one of the quietest places mapped. Overall, employees appear to be most exposed to noise in the Packaging area, followed by time spent at social events, in the Warehouse, and during break at work. Meanwhile, they were the least exposed to noise at home.

Loudness is only one of the factors determining the harmfulness of noise. The other main factor is duration of exposure. In order to get insight into this aspect of exposure, dose computation was performed according to NIOSH standards [13]. This analysis strives to account for peak levels that may have been smoothed out from the average levels due to underrepresentation (e.g., brief exposure to high sound levels). The incremental dose was registered by the noise dosimeter as shown in Figure 5. To get location-wise doses, the hourly increment was computed. The average dose increase was evaluated for each location (Figure 3a) and each population employment subgroup (Figure 3b). Overall, observations are quite similar as for noise levels. The bottling department showed the highest dose increase per hour (+22.7% on average) followed by social events (+11.0%) and the Brewery Canteen. This is logical given the fact that Packaging employees receive a higher average dose per hour.

## 4. Discussion

### 4.1. Personal Protective Equipment

It was necessary to contextualize the recorded noise levels within the brewery’s safety rules and regulations. Therefore, observations and interviews were conducted on-site. Most sites that reported high noise levels within the brewery required mandatory use of personal protective equipment (PPE). Departments such as Brewing, Packaging, and Warehousing required the use of earplugs. The Utilities building required earplugs to be combined with headphones. The latter reported a low noise level as all duties were performed from an insulated control room. PPE was only mandatory for personnel when crossing the engine rooms. On the other hand, some locations reporting high noise levels are currently out of regulatory scope; the Brewery Canteen and Social Events are not restricted due to the personal nature of these locations. Yet, Social Events are the main work environment for a group of employees: sales representatives. In Nigeria, the main duty of these employees is to visit possible sales locations such as Bars, Clubs, and analogous places. Sales employees do not currently fall under any hearing protection regulations. Yet, in the questionnaire, 48.5% of sales employees reported that they considered PPE’s as necessary, while 51.5% also reported not wearing them when recommended. Regular on-site controls result in high compliance on brewery grounds. Yet, we believe more education and information campaigns would likely result in enhanced self-awareness and personal decision-making for all employees regarding hearing protection in- and especially outside the brewery. Ultimately, overall reports of hearing-related injuries were, however, not above normal recurrence (1–2%). In total, 5% of the employees reported having problems hearing, either sometimes, often, or always.

### 4.2. Study Results

Taking the present study into consideration, there are some limitations and possible improvements worth discussing. First, given the local conditions, it was not possible to perform random sampling within the brewery population. No intentional bias was induced but we cannot exclude that some characteristics of the observed sample may differ in terms of unobserved confounders from the total population. For example, employees not willing to participate may already suffer some form of reduced hearing and may be afraid of possible repercussions if diagnosed. Hierarchy and scarcity of employment are deeply rooted problems in the Nigerian working culture. Overall, as is shown by the estimates in Table 1, a relatively representative distributed sample of the population was included. Furthermore, it was logistically not possible to allow every employee to rest one hour prior to the tests and, therefore, performance in audiometry may be reduced due to temporary threshold shift (TTS) [14]. This was also true for employees undergoing the test after their night shift, where tiredness had a potentially large influence on the results, with two reported cases of tests being postponed for this reason. Generally, a study’s logistical procedure should be developed in accordance with local schedules and habits, which cannot be fully anticipated otherwise than without prior on-site explorations. Dosimetry was shown to improve the quality of the measurement. Yet, some further improvements should be implemented to make it a reliable source of information. Logging should be improved, and input should be restricted to a specified set of possibilities to reduce the need for extrapolation of specifically reported data and data loss. In this case, unclear or missing data represented an important part of our recordings and may have influenced the results. It is difficult to judge how far informative missingness might have influenced results, as few comparable studies of this kind have been published. Second, behavioral change while wearing the device should be investigated through a literature survey and taken into account in the results. It should be noted that the lower threshold of the noise dosimeter was 70 dB, all values below this level were not recorded. The value of 35 dB was imputed to in values below the detection limit, explaining the values in the results that are below threshold. Therefore, all values in the table below this threshold must be interpreted with precaution and should, therefore, be considered exploratory analyses. Other options include multiple regression and Tobin regression.

Observations concerning the awareness of employees resulted in a suggestion for better knowledge and information programs. Effort should be put into implementing these into the culture as they may be a solution for what appeared to be one of the main causes of risks around noise [15,16]. As of December 2019, awareness programs and learning tools are more widely used in the brewery.

Since the publication of Ologe et al. [6] it seems the culture within workplace has changed and matured in Nigeria. A significant improvement was observed in terms of noise-induced hearing loss among workers. We have observed a healthy population with an overall average proportion of NIHL of 12.9% (95% CI: 10.1–16.2), which is quite high for western standards but is significantly lower than previously reported cases in sub-Saharan Africa [9,10]. Safety at workplace appeared to be a term that has entered the common language and precautions are taken seriously against the hazards of loud noise. Our subjects indicated that they understood the need for hearing protectors even when these were not obliged by legislation as shown by our subject group of sales employees. Though improvements have already been made in the past decade, many are still required, many have. Moreover, there is fertile ground for the use of training programs within the workforce. Use and effectiveness of these programs could be a great subject to be evaluated in the future.

While not reporting any disconcerting results, this study has given many insights in the populations at risk in the brewery and the way they may be threatened. The setup of this study also taught many lessons about the way these observations could yield better results. This may, therefore, also serve as a pilot for further investigations mainly in terms of risks outside the work environment and a way to raise awareness around the risks of noise in developing countries but also as a reminder for western countries.

## 5. Conclusions

A healthy hearing profile was measured for a large proportion of all brewery employees. NIHL was diagnosed in 12.9% of all participants, primarily employees from the Bottling department (16.7%), Sales (12.9%), and Utilities (11.5%) departments. The prevalence of NIHL increases with age. The prevalence of NIHL found in these three departments gives cause for concern as it is 10% larger than other departments and way above the prevalence in sub-Saharan Africa [17] by a significant degree. Statistical evaluation showed a significant, global influence of department; however, the post hoc analysis could not identify individual difference between said departments in most cases. Only the Bottling department showed a single significant influence on developing NIHL. When looking at the history of each employee, again, only the Bottling department showed a significant effect on hearing loss. 

The departments showing the highest incidence of NIHL were also the ones reporting the highest noise levels in the audiometric recordings. Fortunately, the observed compliance with the auditory protection equipment rules on site was quite high. Follow-up research is necessary to identify any evolution in the hearing profile. Finally, the high level of Sales employees diagnosed with NIHL could be explained by the high levels recorded at the different social events attended for business reported in this part of the study. 

## Figures and Tables

**Figure 1 ijerph-17-02880-f001:**
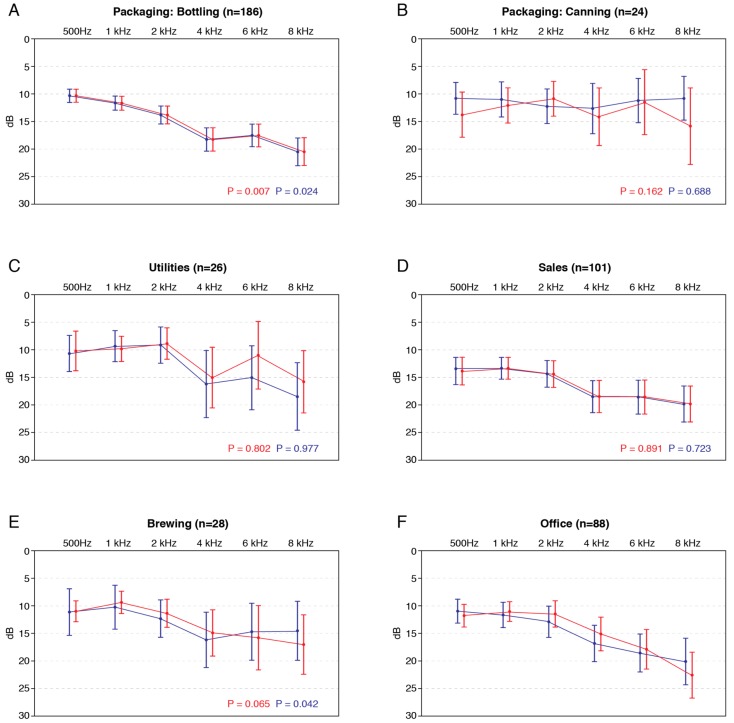
Average air conduction thresholds per department. Both left (blue) and right (red) ear are shown as well as 95% confidence interval. *p* values for unpaired sample T-test are shown per department versus Office for 4 kHz threshold. (**A**) Average ear conduction thresholds for employees working at the bottling department; (**B**) Average ear conduction thresholds for employees working at the canning department; (**C**) Average ear conduction thresholds for employees working at the utilities department; (**D**) Average ear conduction thresholds for employees working as Sales representatives; (**E**) Average ear conduction thresholds for employees working at the brew house (**F**) Average ear conduction thresholds for employees working in the brewery offices.

**Figure 2 ijerph-17-02880-f002:**
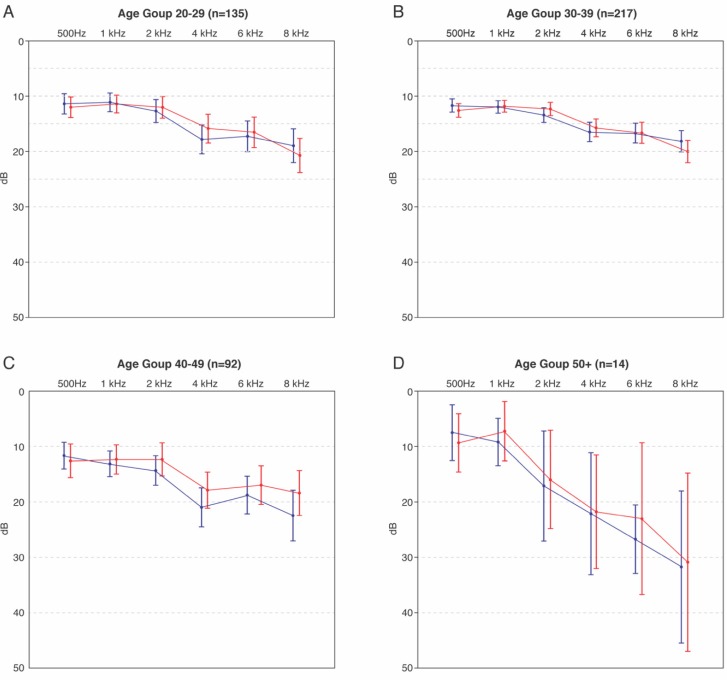
Average air conduction thresholds per age group. Both left (blue) and right (red) ear are shown per group as well as 95% confidence interval. (**A**) Average air conduction thresholds for subjects between 20 and 29 years of age; (**B**) Average air conduction thresholds for subjects between 30 and 39 years of age; (**C**) Average air conduction thresholds for subjects between 40 and 49 years of age; (**D**) Average air conduction thresholds for subjects between 50 years of age and over.

**Figure 3 ijerph-17-02880-f003:**
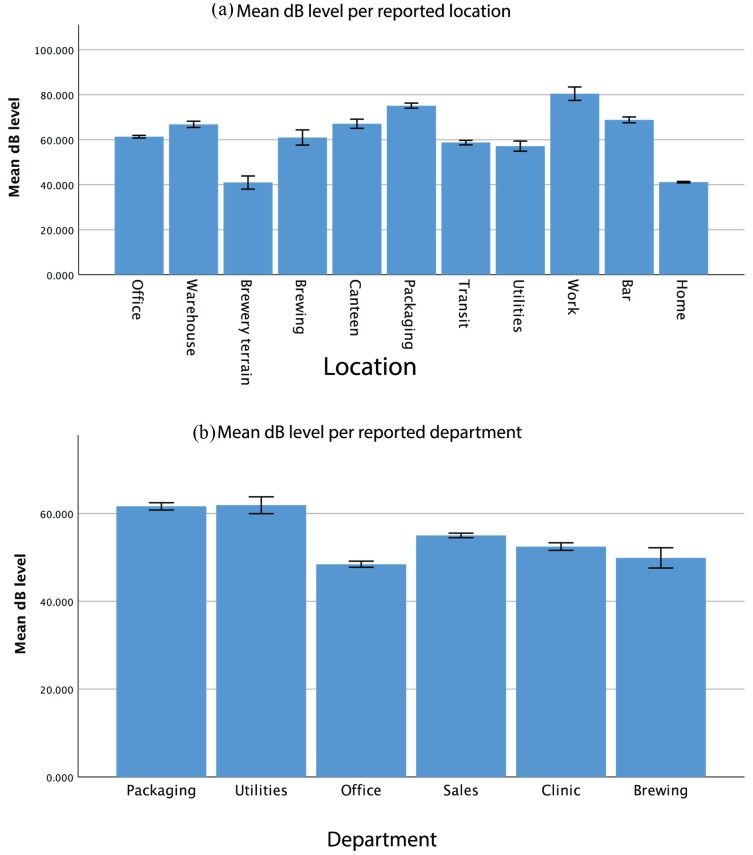
Mean dB levels measured per reported location (**a**) and per reported department (**b**); error bars represent the 95% confidence interval.

**Figure 4 ijerph-17-02880-f004:**
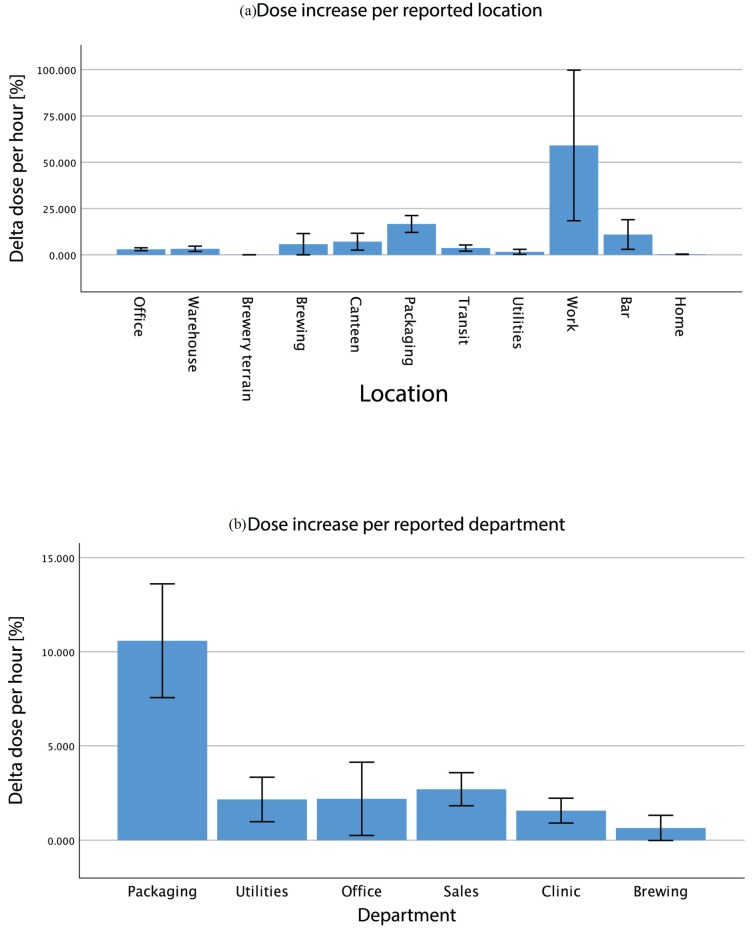
Average dose increase per hour per reported location (**a**) and per reported department (**b**); error bars represent 95% confidence intervals.

**Figure 5 ijerph-17-02880-f005:**
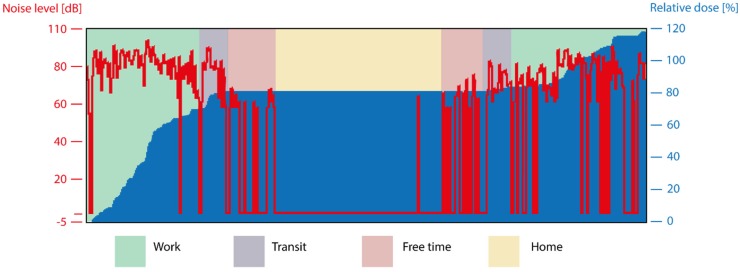
Typical personal noise dosimetry (PND) recording with logbook overlay: Graphical visualization of a 24-h run of a PND. The red line (left axis) is the variation of the recorded noise level over time (horizontal axis 0 to 24 h). Levels below 70 dB were not recorded and showed as drops in the recording. Blue graph (right axis) shows the cumulative dose over time. Colored areas in the background represent the whereabouts of the subject as reported in his/her logbook.

**Table 1 ijerph-17-02880-t001:** Attendance per department.

	Department	*n*	Percent	Est. ^a^
	Packaging: Bottling	186	40.6	164
	Packaging: Canning	24	5.2	28
	Utilities	26	5.7	10
	Office	88	19.2	593
	Sales	101	22.1	173
	Brewing	28	6.1	23
	Warehousing	5	1.1	36
Total		N = 458		1027

a. Estimated employees per department as of December 2019.

**Table 2 ijerph-17-02880-t002:** Age distribution of participants.

		*n* (Male Female)	%
Age	20–29	135 (M = 120 F = 15)	29.5
	30–39	217 (M = 182 F = 34)	47.4
	40–49	92 (M = 84 F = 8)	20.1
	50+	14 (M = 13 F = 1)	3.1
	Total	*N =* 458	100

**Table 3 ijerph-17-02880-t003:** Mean loss around 4 kHz as compared to surrounding frequencies: The difference between the 4 kHz threshold and the surrounding frequencies was determined and separated in two categories: difference less than 10 dB (left column) and difference larger than 10 dB (right column). The distribution for each department between the two categories is shown in each row.

	Difference < 10 dB (95% CI)		Difference > 10 dB (NIHL) (95% CI)	
Bottling	83.3 (77.5-88.2)%	155	16.7 (11.8–22.5)%	31
Canning	95.8 (82.1–99.5)%	23	4.2 (0.5–17.9)%	1
Utilities	88.5 (72.3–96.6)%	23	11.5 (3.4–27.7)%	3
Sales	87.1 (79.6–92.6)%	88	12.9 (7.4–20.4)%	13
Brewing	89.3 (74.1–96.9)%	25	10.7 (3.1–25.9)%	3
Warehousing	100%	5	0.0	0
Office	90.9 (74.1–95.6)%	80	9.1 (4.4–16.4)%	8

**Table 4 ijerph-17-02880-t004:** ANOVA for between-subject effects: The mean difference between the 4 kHz threshold and surrounding frequencies was used in an analysis of variance (ANOVA) to determine any effect of age, gender, or department on this difference. Overall model parameters are shown in the top table. Age yielded a largely significant effect (*p* < 0.05) while gender did not. Office was used as a control as it was expected to be the normally exposed. Overall department yielded a significant effect (*p* < 0.05) but individual effect could not be found.

Source	Test of Between-Subjects Effects			
	df		Sig.	
Corrected Model	**8**		0.000	
Intercept	1		0.143	
Age	1		0.000	
Gender	1		0.201	
Current Position	6		0.014	
Error	447			
Total	456			
Corrected Total	455			
Parameter	Parameter estimates			
	B	Sig.	95% CI for B	
			Lower Bound	Upper Bound
Intercept	−5.922	0.100	−12.983	1.140
Age	0.266	0.000	0.110	0.343
Gender	−1.801	0.201	−4.567	0.965
Packaging: Bottling	3.654	0.007	1.006	6.302
Packaging: Canning	−2.265	0.275	−6.335	1.805
Utilities	3.697	0.073	−0.350	7.745
Sales	2.247	0.104	−0.467	4.962
Brewing	3.263	0.093	−0.547	7.072
Warehousing	−0.407	0.917	−8.083	7.268
Office	0 ^a^			

Dependent Variable: mean difference between 4 kHz threshold and other frequencies. ^a^. This parameter is set to zero because it is redundant.

**Table 5 ijerph-17-02880-t005:** ANOVA regression model: Total time spent per department by each employee was ascertained, including employees who might have switched departments during their time at the brewery. Two models were compared; age, gender, and total time in a department were only added only in the second model. This was meant to correct for the high influence of age on the model parameters as determined in Table 4.

ANOVA ^a^					
Model		df		Sig.	
**1**	Regression	2		0.000 ^b^	
	Residual	453			
	Total	455			
2	Regression	9		0.001 ^c^	
	Residual	446			
	Total	455			
Coefficients					
Model		Unstandardized Coefficients		95% CI for B	
		B	Sig.	Lower Bound	Upper Bound
1	(Constant)	0.608	0.804	−4.203	5.418
	Gender	−3.689	0.002	−6.058	−1.320
	Age	0.166	0.002	0.059	0.272
2	(Constant)	−1.679	0.595	−7.876	4.517
	Gender	−3.002	0.020	−5.531	−0.473
	Age	0.225	0.013	0.047	0.402
	Bottling TO *	0.077	0.456	−0.126	0.281
	Canning TO	−0.423	0.057	−0.858	0.012
	Utilities TO	−0.012	0.943	−0.353	0.328
	Office TO	−0.157	0.233	−0.415	0.101
	Sales TO	−0.241	0.202	−0.611	0.130
	Brewing TO	−0.091	0.511	−0.363	0.181
	Warehousing TO	−0.156	0.666	−0.868	0.555

a. Dependent Variable: mean difference between 4 kHz threshold and other frequencies. b. Predictors: (Constant), Age, Gender. c. Predictors: (Constant), Age, Gender, sales TO, warehouse TO, Canning TO, Utilities TO, Brewing TO, Bottling TO, Office TO. * TO: Time Overall.

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
