# Peer review of "Noise Exposure and Hearing Loss among Brewery Workers in Lagos, Nigeria"

_ijerph, 2020, doi:10.3390/ijerph17082880_

Round 1

Reviewer 1 Report

The article deals with a significant problem, which is often neglected, especially in underdeveloped countries. In accordance with generally accepted principles, an article of a scientific nature should be treated as an article presenting the results of original empirical, theoretical, technical or analytical research, which, in addition to the publication title, authors' surnames and names, presents the current state of knowledge in the described field, the used research methodology and the process research, its results and conclusions contributing to the development of the current state of knowledge.

Admittedly, the article presented for review has all these features but in my opinion it would require additions or even an edition in the part concerning the emphasis on the contribution to the development of existing knowledge. The conclusions presented in the end seem to be not very original and very general. This, in my opinion, may be influenced by a rather cursory treatment of the part related to the analysis of the current state of knowledge in the scope described later in this publication. It is particularly important in connection with the fact that I would include this publication in the so-called case study - a publication which is an analysis of a given case (usually a real one) giving the opportunity to draw conclusions about the causes and results of the case described in it.

The article deals with a significant problem, which is often neglected, especially in underdeveloped countries. In accordance with generally accepted principles, an article of a scientific nature should be treated as an article presenting the results of original empirical, theoretical, technical or analytical research, which, in addition to the publication title, authors' surnames and names, presents the current state of knowledge in the described field, the used research methodology and the process research, its results and conclusions contributing to the development of the current state of knowledge.

Admittedly, the article presented for review has all these features but in my opinion it would require additions or even an edition in the part concerning the emphasis on the contribution to the development of existing knowledge. The conclusions presented in the end seem to be not very original and very general. This, in my opinion, may be influenced by a rather cursory treatment of the part related to the analysis of the current state of knowledge in the scope described later in this publication. It is particularly important in connection with the fact that I would include this publication in the so-called case study - a publication which is an analysis of a given case (usually a real one) giving the opportunity to draw conclusions about the causes and results of the case described in it.

Author Response

Dear Sir or Madam, Dear reviewer,

Thank you for your constructive review. I have tried to address your remarks as best as I could. First and foremost, I have added a section within the introduction to contextualize the paper within the current state of knowledge and highlight its interest for the public. This introduction contains further references within the domain. The state of knowledge gives the article better ground to stand on. In the conclusion, I have tried to bring the findings of our paper a more prominent place within the context introduced in the introduction.

Thank you again, I hope to have addressed your concerns about the paper.

Sincerely,

Nino Wouters 

Reviewer 2 Report

The manuscript from Wouters et al., reports the results from two observational studies on noise-exposure and hearing from a local brewery in Nigeria. The study is of significance and will be of interest to the readers of IJERPH. For the most part, the studies were carried out well and appropriately analyzed. My main comment is on the interpretation of audiometric results presented on page 4. The conclusions  presented are not fully supported by the statistical results presented in table 3. It is not clear from the description or table how ANOVA model was setup. Why does age have only 1 degree of freedom?  A more detailed description of statistics is required.

Some minor comments:

1) For the purpose of clarity, please mention that  frequency weighting A was used for dosimetry.

2) Throughout the manuscript : “give insight in”, “give insight into” is better.

3) line 44: “In the following”. A word is probably missing after following.

4) lines 68-69: “The resulting scale…ANOVA”. Not sure what the authors are trying to convey with this sentence.

5) line 103: including should be corrected to included.

6) line 110: What does noise level recordings refer to here?

7) line 341: may be revision is a better word here.

Author Response

Dear Sir or Madam, Dear reviewer,

Thank you for your encouraging words and constructive remarks. I have tried to address them as best as I could. Here are my thoughts.

First, I want to briefly address your textual comments before entering into details of the content remarks.

  1. As per your recommendation, I have added the sentence: “[…] with frequency weighting A.” (line 203).
  2. The few instances of “insight in” have been replaced with “insight into”.
  3. To clarify the sentence: “In the following we will start […]” (line 44) has been replaced with “In the following paper, we will start […]”.
  4. I have tried to reformulate the sentence on lines 68-69 in a clearer manner: “The resulting threshold values for these frequencies were used as a measure for hearing loss and in combined with the questionnaire results through further analysis by ANOVA and frequency reports.” Hope this makes the intentions clearer.
  5. Corrected to “[…], gender included.”
  6. I assumed you referred to line 113 as the lay-out may have shifted. Anyhow, I replaced “noise levels” with “Hearing Thresholds” (line 113).
  7. Corrected this to “[…] revision phase of this paper.”

As for your remarks concerning the statistical analysis on page 4, I have to admit being a bit shy about these as I am not completely familiar with the subject yet. I have tried to further clarify the thoughts and methods behind the tests. I have tried to address the main concern about age by specifying the fact that this is a scale variable and therefore only has one degree of freedom. Furthermore, the use of the ANOVA has been further clarified by more clearly stating the aim. Conclusion have been slightly changed in order to present a clearer relation with the results.

Thanks again, I hope to have correctly addressed your concerns about the paper.

Sincerely,

Nino Wouters